# Bridging Layout and RTL: Knowledge Distillation based Timing Prediction

Mingjun Wang [* 1 2 3 4]  Yihan Wen [* 5 2]  Bin Sun [1 3]  Jianan Mu [1 3]  Juan Li [5]  Xiaoyi Wang [5]  Jing Ye [1 4]  Bei Yu [2]  Huawei Li [1 3 4]

## Abstract

Accurate and efficient timing prediction at the register-transfer level (RTL) remains a fundamental challenge in electronic design automation (EDA), particularly in striking a balance between accuracy and computational efficiency. While static timing analysis (STA) provides high-fidelity results through comprehensive physical parameters, its computational overhead makes it impractical for rapid design iterations. Conversely, existing RTL-level approaches sacrifice accuracy due to the limited physical information available. We propose RTLDistil, a novel cross-stage knowledge distillation framework that bridges this gap by transferring precise physical characteristics from a layout-aware teacher model (Teacher GNN) to an efficient RTL-level student model (Student GNN), both implemented as graph neural networks (GNNs). RTLDistil efficiently predicts key timing metrics, such as arrival time (AT), and employs a multi-granularity distillation strategy that captures timing-critical features at node, subgraph, and global levels. Experimental results demonstrate that RTLDistil achieves significant improvement in RTL-level timing prediction error reduction, compared to state-of-the-art prediction models. This framework enables accurate early-stage timing prediction, advancing EDA's "left-shift" paradigm while maintaining computational efficiency. Our code and dataset will be publicly available at https://github.com/sklp-eda-lab/RTLDistil.

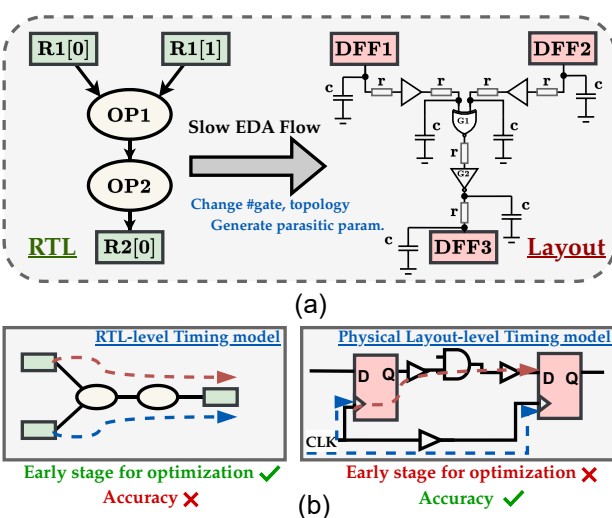

Figure 1: (a) Gap between RTL-level representation and layout-level representation in the chip design process. (b) RTL-level timing models provide faster and earlier predictions compared to the slower and later layout-level models.

## 1. Introduction

Digital circuits constitute the cornerstone of contemporary computing infrastructure (Agarwal & Lang, 2005; Clements, 2006). As illustrated in Figure 1(a), integrated circuit (IC) design involves a complex evolution spanning multiple stages of abstraction and representation paradigms (Chang, 1997; Chen et al., 2024). This progression moves from high-level behavioral descriptions to the topological interconnect of logic cells and ultimately culminates in a transistor- and interconnect-level layout ready for physical manufacturing (Bryant et al., 2001). However, traditional EDA flows often follow a top-down, waterfall-like methodology, where performance bottlenecks typically emerge late in the design cycle, resulting in lengthy verification and iteration periods (Wang et al., 2009). Layout-level timing prediction is too late and too slow. Consequently, recent research efforts have advocated an EDA "left-shift", wherein performance prediction and issue detection are introduced at earlier stages—such as the RTL level—to facilitate more timely design adjustments and optimization of key metrics (Xing, 2024; Zeng, 2024).

---

[*]Equal contribution  [1]State Key Lab of Processors, Institute of Computing Technology, Chinese Academy of Sciences, China [2]The Chinese University of Hong Kong, HKSAR [3]University of Chinese Academy of Sciences, China [4]CASTEST Co., Ltd., China [5]Beijing University of Technology, China. Correspondence to: Jianan Mu <mujianan@ict.ac.cn>, Bei Yu <byu@cse.cuhk.edu.hk>.

*Proceedings of the 42$^{nd}$ International Conference on Machine Learning*, Vancouver, Canada. PMLR 267, 2025. Copyright 2025 by the author(s).

As shown in Figure 1(b), integrating such left-shift strategies not only lowers the cost of late-stage rework but also opens new avenues for pinpointing critical physical characteristics and performance bottlenecks sooner in the design flow (Jiang, 2024; Yang et al., 2022).

However, as in Figure 1(b), despite its efficiency for early-stage optimization, performing accurate performance analysis and prediction at the RTL stage remains challenging (Yang et al., 2015). Existing models, such as RTL-Timer (Fang et al., 2024), estimate timing by synthesizing RTL code without accounting for actual parasitic parameters, producing moderately accurate results. Yet a substantial gap persists between these RTL-based estimates and true post-layout timing. In contrast, layout-level timing analysis inherently incorporates more precise physical details—ranging from each cell's realistic Resistor and Capacitance (RC) parameters to complex interconnect RC models—thus delivering more reliable predictions (Huang et al., 2023). Our experiments indicate that even applying the state-of-the-art RTL model trained end-to-end with layout-level labels yields only around 60% accuracy in layout-level timing predictions.

When compared with predicting the performance of a post-synthesis circuit, the core bottleneck in accurately forecasting layout-level timing lies in the intricate physical parameter distribution—particularly RC parameters—present in the actual circuit netlist (Sait & Youssef, 1999). This distribution critically affects the final timing but is difficult to derive purely from RTL descriptions (Kundert & Zinke, 2005). Consequently, a critical question in early-stage performance prediction arises: how can one effectively extract such underlying physical layout phenomena from the more abstract RTL representation? It is precisely at this juncture that knowledge distillation becomes especially relevant, offering a structured means to transfer these crucial physical insights.

To address these challenges, we propose a knowledge distillation framework, RTLDistil, which bridges the gap between the early-stage RTL and the final layout design. Specifically, at the layout stage, a high-precision teacher model captures realistic physical effects through iterative forward and reverse propagation. Simultaneously, a lightweight student model at the RTL stage assimilates critical timing characteristics, such as arrival time, by drawing on this distilled knowledge. Adopting a cross-stage perspective, RTLDistil employs a multi-granularity alignment strategy (node, subgraph, and global levels) to enhance both precision and efficiency. The primary contributions of RTLDistil can be summarized as follows:

- Cross-Stage Knowledge Distillation. We propose an efficient cross-stage learning approach that transfers

layout-aware timing characteristics to RTL-level predictions. This is a valuable early attempt to apply the concept of knowledge distillation to the EDA domain.

- Multi-Granularity Distillation Learning. We design alignment mechanisms at node, subgraph, and global levels to strengthen the model to capture layout features across different granularities. Each of the three granularities naturally corresponds to specific circuit structures relevant to timing.

- Efficient Forward and Reverse Propagation. We develop a domain-specific asynchronous forward-reverse propagation strategy that balances accuracy with computational efficiency by combining the iterative forward and reverse propagation of the teacher model with a lightweight inference of the student model.

Experimental results on a broad variety of circuits demonstrate that the proposed RTLDistil achieves impressive results compared with state-of-the-art RTL-level timing prediction models. These results validate RTLDistil's effectiveness in enabling accurate early-stage timing prediction, supporting the "left-shift" paradigm in industrial workflows while maintaining computational efficiency.

## 2. Background and Related Work

The research spans several domains, including traditional static timing analysis (STA), RTL-based timing prediction, the application of GNNs in EDA, and knowledge distillation (KD). Below, we summarize the relevant prior works and highlight the core challenges in this field.

**Static Timing Analysis and Timing Prediction**. Static timing analysis (STA) is a critical assessment in chip electronic design automation (EDA) for evaluating whether a circuit satisfies timing constraints (Sapatnekar, 2018; Blaauw et al., 2008). By performing parasitic parameters extraction on the chip physical layout and the timing metrics in the process library, post-layout stage STA calculates precise timing metrics such as the arrival time (AT) at each flip-flop, the circuit worst negative slack (WNS), and the circuit total negative slack (TNS) (Licastro, 2022; Chowdhary et al., 2005). However, analytical STA methods incur heavy runtime due to the complexity of delay propagation calculations (Guo et al., 2024b). Consequently, timing prediction methods have been widely explored in recent research.

Layout-stage timing prediction methods emphasize extracting features from the physical layout and mapping them to the gate-level netlist corresponding to the process library (Guo et al., 2022; Ye et al., 2023). These approaches enable direct prediction of timing metrics with high accuracy. However, they are not suitable for meeting the

requirements of RTL-level prediction. STA at the layout stage depends on a complete mapping to the specific process library and physical layout, making it computationally expensive and impractical for early design stages (Lienig et al., 2020). To support the EDA left-shift paradigm, which advances critical tasks earlier in the design process (Guo et al., 2024a), several studies have explored early-stage timing estimation at the RTL level.

On the other hand, some recent works utilize statistical or simplified models to predict the delay of critical paths to each register based on logical structures (Sengupta et al., 2023; Xu et al., 2022; Lopera et al., 2021a). While these methods enable RTL-level predictions, they exhibit significant inaccuracies due to the absence of physical characteristics, such as capacitance and interconnect delays.

Recent works such as MasterRTL (Fang et al., 2023) and RTL-Timer (Fang et al., 2024) have made progress in RTL-based timing prediction. MasterRTL leverages a bit-level Simple Operator Graph (SOG) and multi-stage machine learning models to estimate TNS and WNS, while RTL-Timer employs a bit-level graph structure with customized loss functions for fine-grained timing predictions at the register and design levels. However, these methods are limited to RTL or gate-level abstractions and fail to effectively incorporate physical characteristics, such as parasitics, interconnect delays, and cell drive strengths (Jariwala, 2011; Kahng et al., 2011). As a result, they struggle to achieve sign-off-level accuracy and provide timing predictions that closely approximate the ground truth of layout-level timing information required in industrial workflows. Moreover, previous works focus on independent paths, which overlook the broader circuit environment, further limiting predictive accuracy.

In conclusion, the existing body of work reveals a fundamental challenge: relying solely on RTL-level logical properties for timing prediction cannot bridge the information gap, and directly depending on STA tools is impractical for rapid early-stage iterations. Consequently, transferring physical characteristics into RTL-level models to enhance prediction accuracy remains an open problem.

**Graph Neural Networks in EDA**. Graph neural networks (GNNs) have emerged as a powerful tool for modeling unstructured data and have been extensively applied in the EDA domain (Ma et al., 2020; Li et al., 2023). In circuit design, circuits are naturally represented as graphs, where nodes correspond to logic elements (e.g., registers, standard cells), and edges represent signal paths (Bairamkulov & Friedman, 2022). GNNs, through graph convolutions and attention mechanisms, can capture rich contextual information between nodes, demonstrating superior performance in tasks such as gate-level network modeling, congestion prediction, and power estimation (Ghose et al., 2021; Sánchez

et al., 2023; Lopera et al., 2021b; Wang et al., 2025).

Some recent works (Fang et al., 2023; Zheng et al., 2024; Lopera & Ecker, 2022) applied GNNs to RTL-level circuit modeling. However, these methods primarily rely on RTL-level features and fail to effectively integrate physical characteristics at the layout stage, overlooking parasitic effects that significantly influence the prediction accuracy. These factors are challenging to model using RTL-level logical abstractions alone. Therefore, leveraging GNNs to combine RTL and layout information across stages remains a critical and underexplored avenue for research.

**Knowledge Distillation**. Knowledge Distillation (KD) is a modeling technique that facilitates the transfer of representational and predictive capabilities from a high-capacity model (teacher) to a simpler counterpart (student) (Hinton, 2015; Zhao et al., 2022; Tang et al., 2020). By guiding the student model to mimic the output distributions of the teacher model, KD enables the student to achieve comparable performance with reduced complexity (Park et al., 2021; Cho & Hariharan, 2019). Traditional KD methods primarily rely on soft targets, where the student learns to approximate the teacher's output logits (Gou et al., 2021; Wang & Yoon, 2021). Recent advancements extend KD to intermediate feature representation, allowing the student model to learn hidden representations from the teacher, further enhancing its capability (Wang et al., 2021).

In circuit design tasks, KD presents unique challenges due to the significant differences in features and dimensionality between RTL and Layout stages. The RTL stage focuses on high-level abstractions, such as logic functionality and registers distribution (Chu, 2006), while the Layout stage includes detailed physical characteristics, such as capacitance, interconnect delay, and parasitic effects (Sherwani, 2012). Transferring high-fidelity physical characteristics from the teacher model to the student model is the key to achieving accurate and efficient timing prediction at the RTL stage. However, designing a KD framework capable of bridging such heterogeneous representations remains a significant challenge.

## 3. Problem Formulation

Given a digital circuit in RTL format, represented as a graph, $G = (V, E)$, where $V$ is the set of register/DFF nodes and the logic nodes, and $E$ represents data flow edges, the goal is to predict the timing-related metrics, such as the arrival time (AT) $\text{AT}_{\text{pred}}(v)$ for each node $v \in V$ at the RTL stage. The ground truth of the arrival time from the layout stage is denoted as $\text{AT}_{\text{label}}(v)$.

The objective is to minimize the error between the predicted and ground truth AT values across all register/DFF nodes,

for example:

$$\text{MAPE}_{\text{AT}} = \frac{1}{|V|} \sum_{v \in V} \left| \frac{\text{AT}_{\text{pred}}(v) - AT_{\text{label}}(v)}{\text{AT}_{\text{label}}(v)} \right|, \quad (1)$$

where $|V|$ is the total number of register/DFF nodes. The output is the predicted arrival time $\text{AT}_{\text{pred}}(v)$ for all register/DFF nodes $v \in V$, aligning with the layout-level timing characteristics.

## 4. RTLDistil Overview

We propose RTLDistil, a dual-model framework comprising a Teacher Model and a Student Model, to address the cross-stage timing prediction problem. The teacher model, operating on the layout-level Netlist Graph, utilizes a high-capacity Graph Neural Network (GNN) to capture complex physical characteristics and generate accurate AT values. The student model, operating on the RTL SOG, employs a lightweight GNN for efficient inference, making it suitable for large-scale RTL-stage prediction.

Figure 2 illustrates the proposed RTLDistil overall workflow. The approach includes four main steps: **A** Graph Construction: Extract the SOG from RTL and the Netlist Graph from Layout, with respective graph features for each stage. **B** Teacher Model Training: Train the teacher model using Layout data as features and labels to produce high-accuracy AT values. **C** Knowledge Distillation: Transfer the physical knowledge encoded in the teacher model to the student model through three levels of granularity, enabling the student to efficiently predict AT at the RTL stage. **D** Student Model Fine Tuning: The student model undergoes further fine-tuning on downstream tasks, such as predicting timing information like AT.

By transferring physical characteristics on layout to the RTL stage, this framework bridges the abstraction gap between RTL and Layout stages, improving prediction accuracy while maintaining computational efficiency.

## 5. Methodology and Model Design

### 5.1. Dual Graphs and Model Architecture

In cross-stage modeling, the RTL and Layout stages differ significantly in information dimensions and features. Table 1 contrasts the features of the Simple Operator Graph (SOG) used at the RTL stage, and the Netlist Graph employed at the Layout stage, highlighting their dimensional and informational differences.

**RTL SOG and the Student Model**. The RTL stage utilizes Verilog/SystemVerilog code to construct a simple operator graph (SOG). SOG nodes represent logical operators (e.g., adders, logic gates, registers) or registers, while

Table 1: RTL SOG vs. layout netlist features.

| Features | Type | Width |
|---|---|---|
| **RTL SOG (Student)** | | |
| SOG cell type | One-hot | 12 |
| Fanout number | Int | 1 |
| Fanin number | Int | 1 |
| Depth Per Input (DPI) | Int | 1 |
| Depth Per Output (DPO) | Int | 1 |
| **Total** | | 16 |
| **Layout Netlist Graph (Teacher)** | | |
| Gate cell type | One-hot | 78 |
| Gate Depth Per Input (DPI) | Int | 1 |
| Gate input pins | Int | 1 |
| Cell drive strength | Int | 1 |
| Fanout capacitance (Rise, Fall) | Float | 2 |
| Fanout resistance | Float | 1 |
| Input slew (4 arcs) | Float | 4 |
| Output slew (4 arcs) | Float | 4 |
| Delay (4 arcs) | Float | 4 |
| **Total** | | 96 |

edges describe data flow relationships. Each node has a 16-dimensional feature vector, including operator types, fan-in/out characteristics, and operators' depth per input (DPI)/depth per output (DPO) values.

The student model is a lightweight graph neural network (GNN), as a multi-head attention-based graph attention network (GAT), which generates 128-dimensional embeddings for each node. To ensure scalability, the student model only requires two rounds of forward-reverse asynchronous propagation to predict node-level timing metrics, such as arrival time (AT) and worst negative slack (WNS), while producing low-dimensional embeddings $\text{embedding}_S(\text{DFF}_i)$ for distillation.

**Layout Netlist Graph and the Teacher Model**. The Layout stage employs a Netlist Graph, and nodes carry 96-dimensional physical features, including drive strength, capacitance, parasitic resistance, capacitance, and LUT delays.

The teacher model is a bigger GAT that produces 512-dimensional embeddings and performs three rounds of forward-reverse asynchronous propagation to simulate realistic STA-like long-path accumulation while seeing a broader view of the surrounding circuitry. It outputs high-precision timing values while generating high-dimensional embeddings $\text{embedding}_T(\text{DFF}_i)$ for knowledge distillation.

### 5.2. Forward-Reverse Propagation Strategy

To imitate the delay propagation of signals in a circuit, both the teacher and student models utilize asynchronous forward-reverse propagation to update node representations, which capture the neighboring structure in the fanin and fanout cones, which is illustrated by Figure 3. The forward

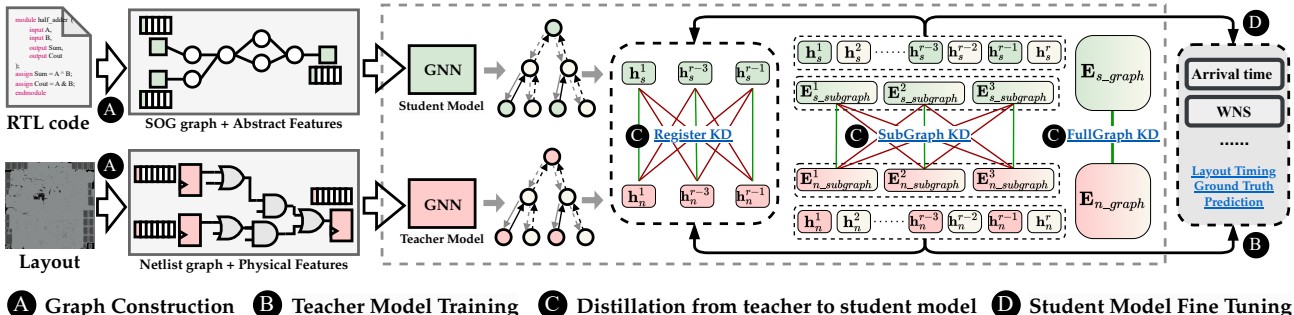

**Ⓐ** Graph Construction   **Ⓑ** Teacher Model Training   **Ⓒ** Distillation from teacher to student model   **Ⓓ** Student Model Fine Tuning

Figure 2: Overview of the RTLDistil framework. The teacher model operates on the layout netlist graph to generate accurate physical timing metrics, while the student model operates on the RTL SOG for efficient inference. Knowledge distillation transfers timing information from the teacher to the student.

propagation update for node $v$ at $(i + 1)$-depth per input node is formulated as Eq. (2a):

$$\mathbf{h}_v^{(i+1)} = \sigma \left( \sum_{u \in \mathcal{N}(v)} \alpha_{uv}^{For} W^{For} \mathbf{h}_u^{(i)} \right), \qquad (2a)$$

$$\mathbf{h}_v^{(j+1)} = \sigma \left( \sum_{u \in \mathcal{R}(v)} \alpha_{uv}^{Re} W^{Re} \mathbf{h}_u^{(j)} \right), \qquad (2b)$$

where $\mathcal{N}(v)$ represents the forward neighboring nodes of node $v$; $h$ denotes the feature vector of nodes; $\mathbf{W}$ denotes the linear transform weight and $\alpha$ denotes the coefficients calculated by the attention mechanism in (Velickovic et al., 2017), where For means forward and Re means reverse. Similarly, the reverse propagation update for node $v$ at $(j + 1)$-depth per output node is formulated as Eq. (2b), where $\mathcal{R}(v)$ denotes the reverse neighboring nodes of $v$.

**Teacher Model**. By leveraging complete physical features, the Teacher executes $\geq 2$ rounds of forward-reverse propagation. The forward pass accumulates path delays, while the reverse propagation spreads information about the surrounding circuitry situation, thus achieving timing analysis accuracy that is closer to the real physical layout.

**Student Model**. To ensure computational efficiency, the Student performs only 2 propagation rounds for rapid node-level timing estimation. Despite fewer iterations, the Student compensates for its limited propagation through knowledge distillation by learning contextual information from the Teacher.

We actually emphasize domain-specific asynchronous forward-reverse propagation for iterative feedback from the sink node back to the source node, capturing RC parasitic and register slack constraints that typical GNNs cannot directly encode. This strategy achieves high accuracy with the Teacher while maintaining the Student's efficiency, en-

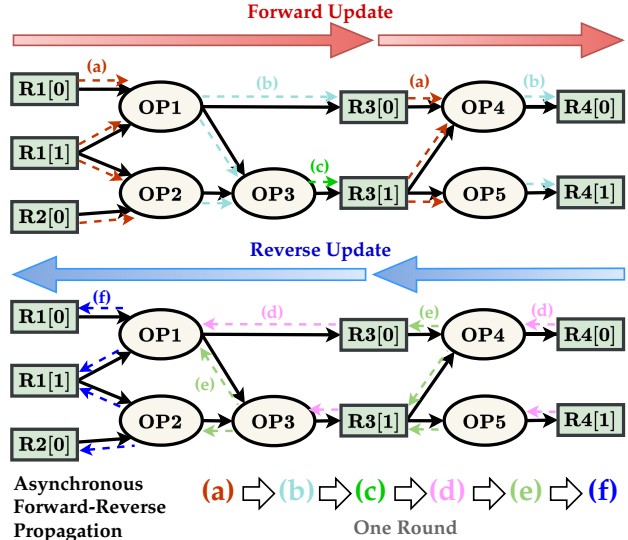

Figure 3: Asynchronous forward-reverse propagation strategy. The RTL SOG is used as an example to show a one-round flow.

suring effective representation alignment during distillation. This mechanism proved crucial to bridging the gap between abstract RTL data and high-accuracy physical layout-level timing insights.

### 5.3. Multi-Granularity Knowledge Distillation

Multi-granularity knowledge distillation (KD) is a cornerstone of the proposed RTLDistil framework for cross-stage timing prediction. It facilitates the transfer of physical knowledge from the teacher model to the student model across multiple granularity levels relevant to timing—node-level, subgraph-level, and global-level—enabling the student to approximate the teacher's high-dimensional physical characteristics effectively and capture timing-critical features at multiple scopes. This hierarchical strategy ensures

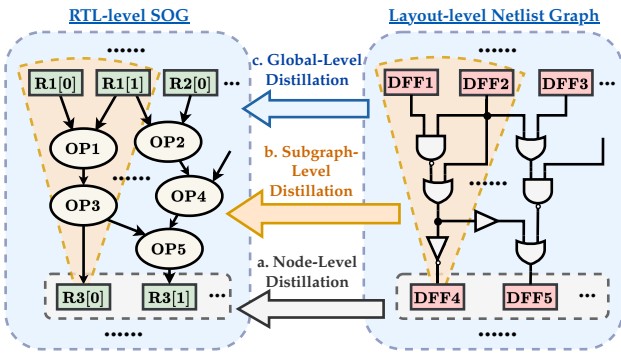

Figure 4: Multi-granularity knowledge distillation framework. Knowledge is hierarchically transferred from the Teacher to the Student across three levels: (a) Node-Level, (b) Subgraph-Level, and (c) Global-Level Distillation. The RTL-level SOG and Layout-level Netlist Graph are aligned to enable accurate cross-stage timing prediction.

that the student model comprehensively learns physical timing dependencies, capturing both local and global circuit behaviors to predict AT, WNS, and TNS robustly during the RTL stage.

Traditional KD methods align the output distributions of a large teacher model and a lightweight Student model by minimizing their divergence. However, in cross-stage timing prediction, the significant differences between RTL and Layout feature spaces render output-only alignment insufficient. To address this, we design a multi-granularity KD strategy that hierarchically transfers knowledge from the Teacher to the Student, ensuring comprehensive learning of physical characteristics across granularities, as illustrated in Figure 4.

**Node-Level Distillation**. Node-level distillation focuses on aligning the feature representations of individual register nodes (*DFF-level*) to ensure that the Student accurately learns the timing properties of each register/DFF. For a register/DFF node $v$, the Teacher and Student generate embeddings $\text{embedding}_T(v)$ and $\text{embedding}_S(v)$. These embeddings are directly compared using the smooth $L_1$ loss, defined as:

$$L_{\text{Reg}} = \frac{1}{|V|} \sum_{v \in V} \text{smooth}_{L_1}\left(\text{embedding}_T(v),\right.$$
$$\left.\text{MLP}_{\text{align}}\left(\text{embedding}_S(v)\right)\right), \quad (3)$$

where $|V|$ is the total number of registers/DFF nodes, and the smooth $L_1$ loss is given by:

$$\text{smooth}_{L_1}(x, y) = \begin{cases} 0.5(x-y)^2, & \text{if } |x-y| < 1, \\ |x-y| - 0.5, & \text{otherwise.} \end{cases} \quad (4)$$

This loss ensures that the Student Model's feature representations align closely with those of the Teacher for each register/DFF node. At the same time, to align the dimensionality between teacher and student embeddings, we employ a two-layer MLP transformation $\text{MLP}_{\text{align}}$ that projects the student's 128-dimensional embeddings to the same 512-dimensional space as the teacher's. This alignment network uses ReLU activation and maintains the semantic information while expanding the feature space.

**Subgraph-Level Distillation**. While node-level distillation aligns single-register features, it may overlook the influence of surrounding logical structures. Subgraph-level distillation addresses this by modeling the fan-in cone of each register, capturing its contextual logic depth, operation types, and timing dependencies.

For a register/DFF node $v$, the fan-in cone subgraph is denoted as $\mathcal{G}_{sub}(v)$, representing the sub-circuit encompassing all combinational logic between $v$ and its contributing registers within a single clock cycle. The fan-in cone of a target register $v$ is determined by traversing the RTL SOG or netlist graph forward from node $v$, collecting all logic gates and connections that influence node $v$, until the traversal terminates at other registers/DFFs. It serves as an intermediate granularity that bridges the gap between fine-grained node-level features and the global circuit graph, enhancing model adaptability for various timing prediction tasks.

For subgraph-level distillation, the Teacher and Student generate subgraph embeddings $\text{embedding}_T(\mathcal{G}_{sub}(v))$ and $\text{embedding}_S(\mathcal{G}_{sub}(v))$. These embeddings are aggregated using mean pooling over the nodes in $\mathcal{G}_{sub}(v)$ and aligned using the smooth $L_1$ loss:

$$L_{\text{subgraph}} = \frac{1}{|V|} \sum_{v \in V} \text{smooth}_{L_1}\left(\text{embedding}_T(\mathcal{G}_{sub}(v)),\right.$$
$$\left.\text{MLP}_{\text{align}}\left(\text{embedding}_S(\mathcal{G}_{sub}(v))\right)\right). \quad (5)$$

This loss encourages the Student to capture the logic and timing interactions within each register's fan-in cone and to focus more on the local timing-relevant information related to each register.

**Global-Level Distillation**. Global-level distillation aligns the representations of the entire circuit to ensure that the Student learns global timing distributions and long-term dependencies. This strategy is particularly effective in correcting biases induced by variations in circuit scale and structure.

The Teacher and Student generate global embeddings $\text{embedding}_T(\mathcal{G})$ and $\text{embedding}_S(\mathcal{G})$ for the entire circuit graph $\mathcal{G}$ using mean pooling. The global embeddings are aligned with the smooth $L_1$ loss:

$$L_{\text{global}} = \text{smooth}_{L_1}\left(\text{embedding}_T(\mathcal{G}),\right.$$
$$\left.\text{MLP}_{\text{align}}\left(\text{embedding}_S(\mathcal{G})\right)\right). \quad (6)$$

By minimizing $L_{\text{global}}$, the Student aligns with the Teacher's global characteristics, improving its ability to capture critical paths across the circuit.

**Integrated Distillation Loss**. The total distillation loss combines the contributions of node-level, subgraph-level, and global-level losses, along with supervised timing prediction losses:

$$L_{\text{total}} = L_{\text{supervised}}(\text{AT}) + \alpha L_{\text{Reg}} + \beta L_{\text{subgraph}} + \gamma L_{\text{global}}, \quad (7)$$

where $L_{\text{supervised}}(\text{AT})$ ensures that the Student's predictions for arrival time (AT) align with ground truth, and the weights $\alpha, \beta, \gamma$ are hyperparameters controlling the relative importance of each distillation granularity. They are tuned based on grid search for optimal performance.

Through iterative gradient descent, the Student progressively aligns its feature representations and timing predictions with the Teacher.

# 6. Experiments and Implementation Details

## 6.1. Experimental Settings

**Datasets**. To enable a comprehensive evaluation of scalability and adaptability, we collected 2004 RTL designs with diverse functionalities and complexities sourced from platforms including GitHub, Hugging Face, OpenCore, and RISC-V projects to reflect real-world industrial needs, including small arithmetic blocks, DSP modules, RISC-V subsystems, etc. This makes the prediction task harder, but we think it's more practical, pervasive, industrially valuable, and closer to the needs of actual industrial processes. In constructing our dataset, we designed a unified, fully-automated back-end flow using state-of-the-art commercial tools—Synopsys Design Compiler (DC) and Cadence Innovus—with a consistent set of optimization switches (e.g., gate sizing, buffer insertion, cell movement, etc.). However, the circuits in our dataset were not finalized under a single fixed configuration. For each design and each back-end optimization-related parameter (e.g., density thresholds, routing constraints, clock constraints), we automatically tried multiple sets of values and iteratively explored multiple different configurations, often conducting tens of design runs, until the circuit reached a state where:

- Placement density no longer increased, and

- Timing metrics converged stably through repeated optimization.

This convergence point serves as a practical proxy for physical design quality, reflecting an optimization level comparable to that of manually refined industrial flows. By doing so, we avoid biasing our dataset toward a singular

"super-convergent" setting and instead generate diverse yet high-quality layouts that are more representative of industrial standards. Our approach reflects a robust and converged implementation quality, providing a meaningful basis for our timing prediction framework.

Importantly, this means our model is not tuned to predict timing under a specific tool configuration, but rather aims to approximate the best achievable timing performance after realistic optimization—an objective more aligned with industrial design goals.

For dataset splits, circuits are split into 80% for training, 10% for validation, and 10% for testing, ensuring a fair evaluation of the model's ability to generalize across different circuits.

**Evaluation Metrics**. We evaluated timing predictions for arrival time (AT), worst negative slack (WNS), and total negative slack (TNS) using three standard metrics: (1) Pearson correlation coefficient (PCC), which assesses the linear correlation between predictions and ground truth; (2) Coefficient of determination ($R^2$), which measures the proportion of variance explained by the model; (3) Mean absolute percentage error (MAPE), which Quantifies prediction error as a percentage of ground truth, with lower values indicating better performance.

**Platform Configuration**. Experiments were conducted on $8 \times$ NVIDIA A100 GPUs, and models were implemented using PyTorch and PyTorch Geometric (PyG). The optimization employed the Adam optimizer with an initial learning rate of $2 \times 10^{-4}$ and batch sizes of 8. Multi-granularity knowledge distillation used grid-searched weights for node-level ($\alpha$), subgraph-level ($\beta$), and global-level ($\gamma$) distillation. With the change of loss weights, we observed small fluctuations across different metrics and believed that the optimal balance varies depending on the data properties, circuit complexity, and the focus of the task objective. Through a coarse-grained grid search, we find equal weights ($\alpha = \beta = \gamma$) that exhibit superior multi-task average performance, thereby clarifying the ablation experiments.

## 6.2. Comparison with Existing Models

We compared RTLDistil against two state-of-the-art (SOTA) models, MasterRTL (Fang et al., 2023) and RTL-Timer (Fang et al., 2024), which represent current advancements in RTL-based timing prediction. We used their open-source code for comparison. Table 2 lists the comparison results. RTLDistil demonstrates consistent superiority over the previous SOTA methods across all metrics.

Specifically, as shown in Table 2, for AT, RTLDistil achieves a PCC of 0.9227, significantly surpassing MasterRTL (0.3498) and RTL-Timer (0.8782). Additionally, RTLDistil

Table 2: Comparison of MasterRTL, RTL-Timer, and RTLDistil on timing prediction tasks.

| Model | Arrival Time (AT) | | | Worst Negative Slack (WNS) | | | Total Negative Slack (TNS) | | |
|---|---|---|---|---|---|---|---|---|---|
| | PCC | $R^2$ | MAPE | PCC | $R^2$ | MAPE | PCC | $R^2$ | MAPE |
| MasterRTL (Fang et al., 2023) | 0.3498 | -0.8718 | 81.20% | 0.7381 | 0.5161 | 61.88% | 0.6255 | -0.2967 | 65.26% |
| RTL-Timer (Fang et al., 2024) | 0.8782 | 0.7568 | 23.39% | 0.8812 | 0.7596 | 40.55% | 0.8451 | 0.6114 | 40.31% |
| RTLDistil (Full Model) | **0.9227** | **0.8486** | **16.87%** | **0.9066** | **0.8141** | **31.37%** | **0.9586** | **0.9181** | **37.95%** |

MasterRTL and RTL-Timer are previous state-of-the-art (SOTA) models. RTLDistil outperforms both in all timing metrics.

Table 3: Performance of Layout Teacher Model, RTL Student Model–RTLDistil (without Distillation, without Fine Tuning, and Full Model with Distillation & Fine Tuning).

| Model | Arrival Time (AT) | | | Worst Negative Slack (WNS) | | | Total Negative Slack (TNS) | | |
|---|---|---|---|---|---|---|---|---|---|
| | PCC | $R^2$ | MAPE | PCC | $R^2$ | MAPE | PCC | $R^2$ | MAPE |
| Layout Teacher Model | 0.9797 | 0.9583 | 11.20% | 0.9580 | 0.9112 | 19.87% | 0.9901 | 0.9802 | 21.86% |
| RTLDistil (w/o Distillation) | 0.8787 | 0.7554 | 22.04% | 0.8658 | 0.6919 | 37.37% | 0.9100 | 0.8216 | 40.31% |
| RTLDistil (w/o Fine Tuning) | 0.9107 | 0.8231 | 17.77% | 0.8874 | 0.7849 | 32.84% | 0.9468 | 0.8955 | **33.30%** |
| RTLDistil (Full Model) | **0.9227** | **0.8486** | **16.87%** | **0.9066** | **0.8141** | **31.37%** | **0.9586** | **0.9181** | 37.95% |

RTLDistil (w/o Distillation) refers to RTLDistil trained without knowledge distillation.
RTLDistil (w/o Fine Tuning) refers to RTLDistil trained after knowledge distillation but without Fine Tuning.

reduces the MAPE to 16.87%. For WNS and TNS, RTLDistil achieves PCC values of 0.9066 and 0.9586, outperforming RTL-Timer by 2.88% and 11.35%, respectively. The reductions in MAPE across WNS and TNS further emphasize RTLDistil's ability to closely approximate layout-level timing.

This consistent improvement across all metrics demonstrates the efficacy of the multi-granularity knowledge distillation strategy in bridging the abstraction gap between RTL and layout stages, enabling accurate and reliable timing predictions at the RTL level. The RTLDistil's high performance is sufficient for early-stage RTL optimization, allowing the design flow to shift left.

**Teacher-Student Model Evaluation**. We also conducted a high-capacity Layout Teacher Model trained on physical features and an RTL Student Model trained without knowledge distillation to analyze the role of multi-granularity knowledge transfer.

Table 3 compares the performance of the Layout Teacher Model, the RTL Student Model—RTLDistil (without Distillation, without Fine Tuning, and Full Model with Distillation & Fine Tuning), offering insights into the effectiveness of the proposed knowledge distillation framework. The Layout Teacher Model achieves the highest performance across all metrics, achieving a PCC of 0.9797 and a MAPE of 11.20% for AT, thereby establishing an upper bound on RTLDistil's achievable accuracy. In contrast, RTLDistil without Distillation, which lacks knowledge distillation, exhibits significantly lower accuracy, with an AT MAPE of 22.04%. This result highlights the inherent limitations of relying solely on RTL-level features to predict layout-level

timing. RTLDistil without Fine Tuning demonstrates a substantial improvement over RTLDistil without Distillation, reducing the AT MAPE to 17.77%, which underscores the effectiveness of the knowledge transfer from the teacher model. Further downstream fine-tuning enhances RTLDistil's accuracy, achieving a PCC of 0.9227 and an MAPE of 16.87% for AT. These results confirm that RTLDistil can closely approximate the teacher model's performance while maintaining computational efficiency, demonstrating its practicality for early-stage timing prediction and optimization.

### 6.3. Ablation Study

**Ablation Study on Multi-Granularity Distillation**. Table 4 compares each ablation setting with a baseline that excludes all distillation. The RTLDistil (Full Model) configuration, which jointly employs node-, subgraph-, and global-level distillation, achieves the highest correlation gains in Arrival Time (AT), Worst Negative Slack (WNS), and Total Negative Slack (TNS). Specifically, it boosts $\Delta$PCC by up to 0.0486, $\Delta R^2$ by up to 0.1222, and consistently lowers $\Delta$MAPE across the three metrics. A closer inspection of the partial variants clarifies the role of each distillation level. Node-level distillation alone captures fine-grained register features, ignoring the global information, occasionally producing large $\Delta$MAPE improvements (e.g., $-5.64\%$ on AT, $-8.53\%$ on WNS), but it yields weaker performance in correlation metrics (PCC, R) for AT, WNS, and TNS overall than the full model, not giving the best results. Incorporating subgraph-level information further exploits local fan-in cones, improving contextual sensitivity. Meanwhile, global-level distillation refines circuit-wide alignment, preventing

Table 4: Ablation study on RTLDistil components, measured as the difference in performance relative to the RTLDistil model without distillation.

| Ablation Configuration | Arrival Time (AT) | | | Worst Negative Slack (WNS) | | | Total Negative Slack (TNS) | | |
|---|---|---|---|---|---|---|---|---|---|
| | $\Delta$PCC | $\Delta R^2$ | $\Delta$MAPE | $\Delta$PCC | $\Delta R^2$ | $\Delta$MAPE | $\Delta$PCC | $\Delta R^2$ | $\Delta$MAPE |
| RTLDistil (Full Model) | **0.0440** | **0.0932** | -5.17% | **0.0480** | **0.1222** | -6.00% | **0.0486** | **0.0965** | -2.36% |
| RTLDistil (w/ Node) | -0.0038 | 0.0750 | **-5.64%** | 0.0195 | 0.0413 | **-8.53%** | 0.0022 | 0.0046 | -6.90% |
| RTLDistil (w/ Node & Global) | 0.0254 | 0.0710 | -5.42% | 0.0336 | 0.0946 | -7.30% | 0.0300 | 0.0410 | **-10.19%** |
| RTLDistil (w/ Subgraph & Global) | 0.0071 | 0.0339 | -4.35% | 0.0204 | 0.0655 | -5.51% | 0.0105 | 0.0134 | -8.70% |

Ablation settings measure the performance difference ($\Delta$) between each configuration and the RTLDistil model without distillation. Node, Subgraph, and Global are respectively shorthand for node-level, subgraph-level, and global-level distillation. Positive $\Delta$PCC and $\Delta R^2$ indicate improved performance, while negative $\Delta$MAPE indicates reduced error.

biases that purely local methods may overlook. Consequently, partial combinations like (w/ Node & Global) or (w/ Subgraph & Global) often enhance $\Delta$MAPE in TNS more aggressively but do not achieve the strong correlation improvements delivered by the full multi-granularity approach.

These observations highlight that simultaneously capturing localized node-level timing details, subgraph-level contexts, and holistic circuit characteristics offers the best balance between reduced timing error and improved predictive correlation at the RTL stage.

**Analysis of Multi-Granularity Distillation Losses**. Figure 5 illustrates the training dynamics of the multi-granularity knowledge distillation process, showcasing the hierarchical alignment between the teacher and student models. The total distillation loss ($L_{\text{total}}$) steadily declines and converges, indicating successful overall alignment. The node-level distillation loss ($L_{\text{Reg}}$) decreases rapidly and stabilizes early, reflecting the efficient alignment of local register-level features, while the subgraph-level distillation loss ($L_{\text{subgraph}}$) declines more gradually, capturing contextual relationships within fan-in cones. The global-level distillation loss ($L_{\text{global}}$) starts higher due to the complexity of global timing distributions but steadily reduces, demonstrating the model's ability to capture overall circuit information. These results validate RTLDistil's hierarchical strategy, where multi-granularity alignment ensures comprehensive learning of timing dependencies and progressively improves prediction accuracy.

The experimental results comprehensively demonstrate that RTLDistil achieves state-of-the-art performance in timing prediction at the RTL stage. By leveraging multi-granularity knowledge distillation, RTLDistil successfully bridges the abstraction gap between RTL and layout stages, achieving high prediction accuracy. These findings confirm RTLDistil's potential as a practical solution for early-stage timing optimization in industrial EDA workflows.

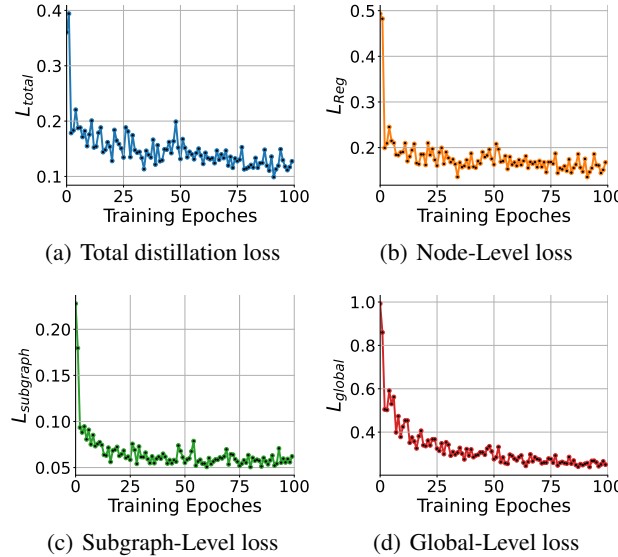

(a) Total distillation loss      (b) Node-Level loss

(c) Subgraph-Level loss      (d) Global-Level loss

Figure 5: Losses of multi-granularity knowledge distillation.

# 7. Conclusion

This paper presents RTLDistil, a cross-stage knowledge distillation framework that bridges the abstraction gap between RTL and Layout stages by transferring high-precision physical characteristics from a layout stage Teacher Model to an RTL Student Model. Through multi-granularity distillation at the node, subgraph, and global levels, the framework achieves layout-level accuracy in timing prediction while ensuring high inference efficiency. The proposed method significantly improved with less MAPE and larger $R^2$ than the SOTA models in timing evaluation, making it suitable for early-stage design optimization. Future extensions include scaling to large SoC designs, incorporating multi-clock domain constraints, and integrating multi-objective optimization for power and area, thereby further enhancing its applicability to industrial chip design workflows. We believe RTLDistil is a valuable early attempt to apply the concept of knowledge distillation to the EDA domain and contributes potentially to AI4EDA and EDA's "left-shift".

## Acknowledgement

This paper is supported in part by the Chinese Academy of Sciences under grant No. XDB0660103, and in part by the National Natural Science Foundation of China (NSFC) under grant No.( 62090024).

## Impact Statement

This paper presents work whose goal is to advance the field of Machine Learning. There are many potential societal consequences of our work, none which we feel must be specifically highlighted here.

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

## A. Appendix: Performance Correlation and Comparative Analysis

This appendix section provides a detailed analysis of the performance of different models for timing prediction, specifically focusing on Arrival Time (AT), Worst Negative Slack (WNS), and Total Negative Slack (TNS). The correlation plots in Figures 6, 7, and 8 illustrate the relationship between the predicted values and the ground truth for each metric. These plots compare the results of MasterRTL, RTL-Timer, RTLDistil without distillation, and RTLDistil with distillation. The red solid line represents the ideal $1:1$ correlation, where the predicted values perfectly match the ground truth, while the dashed black lines mark the $3\sigma$ confidence boundaries, which provide a statistical measure to evaluate the consistency and reliability of predictions by identifying significant deviations. The evaluation highlights the improvements brought by the RTLDistil framework, particularly after incorporating the multi-granularity knowledge distillation process.

### A.1. Arrival Time (AT) Correlation Analysis

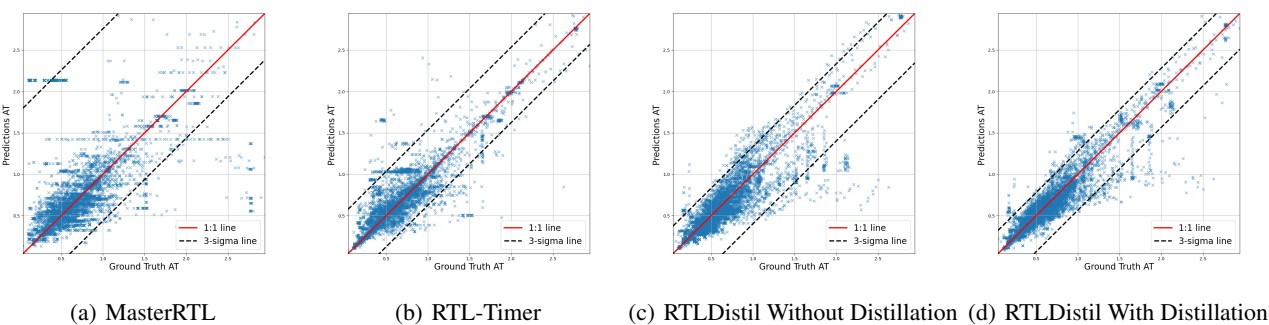

| (a) MasterRTL | (b) RTL-Timer | (c) RTLDistil Without Distillation | (d) RTLDistil With Distillation |

Figure 6: Arrival Time (AT) correlation plots for different models. The proposed RTLDistil with distillation exhibits the best alignment with ground truth AT values.

Figure 6 presents the correlation plots for Arrival Time (AT) predictions across four models: MasterRTL, RTL-Timer, RTLDistil without distillation, and RTLDistil with distillation. The red solid line represents the ideal $1:1$ correlation, while the dashed black lines mark the $3\sigma$ confidence boundaries. These plots highlight the progressive improvements achieved by RTLDistil in aligning predictions with ground truth AT values.

In Figure 6(a), MasterRTL exhibits significant scatter, with many predictions deviating beyond the $3\sigma$ boundary, indicating poor alignment due to its inability to model physical characteristics effectively. Figure 6(b) shows that RTL-Timer improves correlation but still suffers from noticeable outliers, reflecting limited accuracy in capturing timing dependencies.

Figure 6(c) demonstrates that RTLDistil without distillation further reduces scatter, achieving better alignment with the $1:1$ line. However, some deviations persist, highlighting the need for more robust physical information transfer. Finally, Figure 6(d) shows that RTLDistil with distillation achieves the best alignment, with predictions tightly clustered along the $1:1$ line and well within the $3\sigma$ boundary. This improvement is attributed to the multi-granularity knowledge distillation strategy, which transfers physical timing characteristics from the teacher model to the student model, effectively bridging the abstraction gap between RTL and layout stages.

In summary, RTLDistil with distillation significantly outperforms prior methods, demonstrating its ability to achieve highly accurate AT predictions by capturing surrounding circuit information and leveraging layout-level knowledge.

### A.2. Worst Negative Slack (WNS) Correlation Analysis

Figure 7 illustrates the correlation plots for Worst Negative Slack (WNS) predictions across four models: MasterRTL, RTL-Timer, RTLDistil without distillation, and RTLDistil with distillation. The red solid line represents the ideal $1:1$ correlation, while the dashed black lines indicate the $3\sigma$ confidence boundaries. These plots evaluate how well each model predicts critical timing violations in a circuit.

In Figure 7(a), MasterRTL shows poor alignment with ground truth WNS values, characterized by significant scatter and numerous points falling outside the $3\sigma$ boundary. This reflects the model's inability to capture critical physical timing

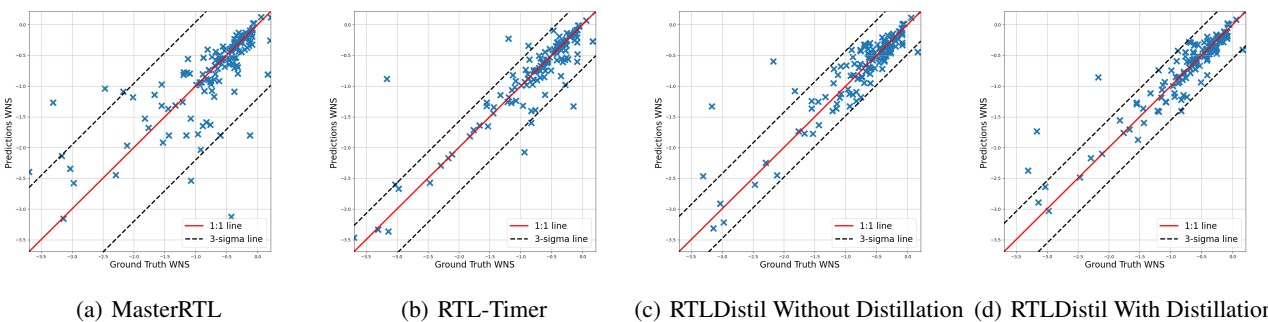

    (a) MasterRTL        (b) RTL-Timer     (c) RTLDistil Without Distillation  (d) RTLDistil With Distillation

Figure 7: Worst Negative Slack (WNS) correlation plots for different models. The proposed RTLDistil with distillation achieves the closest alignment with ground truth WNS values.

dependencies due to its reliance on logical abstractions. Figure 7(b) demonstrates that RTL-Timer improves over MasterRTL by reducing scatter and outliers, achieving moderate alignment with the $1 : 1$ line. However, it still struggles to accurately predict severe timing violations.

Figure 7(c) presents the performance of RTLDistil without distillation, which further reduces the number of outliers and achieves better alignment with the $1 : 1$ line. While this indicates improved predictive capability, some deviations persist due to the lack of physical knowledge transfer. Finally, Figure 7(d) shows that RTLDistil with distillation achieves the closest alignment, with predictions tightly clustered along the $1 : 1$ line and most points well within the $3\sigma$ boundary. This highlights the effectiveness of the multi-granularity knowledge distillation framework in transferring critical physical characteristics from the teacher model to the student model, enabling precise WNS predictions.

In summary, RTLDistil with distillation outperforms all prior methods, demonstrating its ability to accurately predict WNS by bridging the abstraction gap and effectively incorporating layout-level timing information into RTL-stage models.

### A.3. Total Negative Slack (TNS) Correlation Analysis

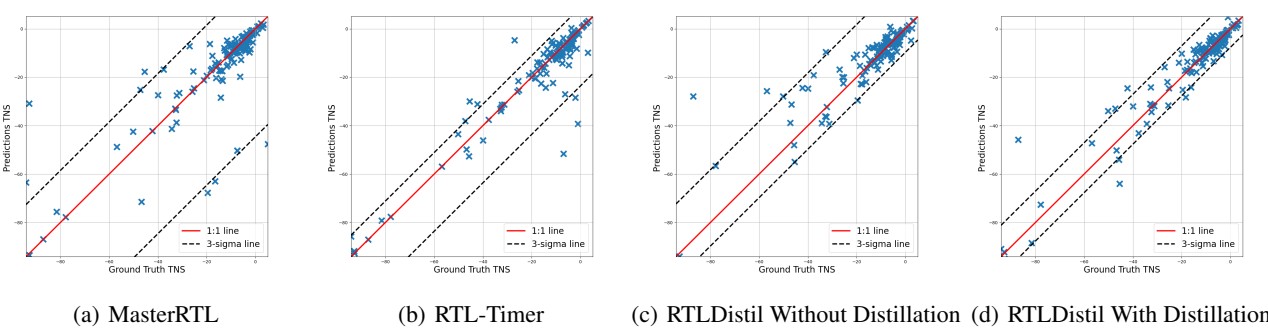

    (a) MasterRTL        (b) RTL-Timer     (c) RTLDistil Without Distillation  (d) RTLDistil With Distillation

Figure 8: Total Negative Slack (TNS) correlation plots for different models. The proposed RTLDistil with distillation achieves the best alignment with ground truth TNS values.

Figure 8 illustrates the correlation plots for Total Negative Slack (TNS) predictions, which quantify the aggregate severity of timing violations in a circuit. The red solid line represents the ideal $1 : 1$ correlation, while the dashed black lines indicate the $3\sigma$ confidence boundaries. These plots reveal the progressive improvements achieved by RTLDistil in predicting TNS values accurately.

In Figure 8(a), MasterRTL performs poorly, with significant scatter and numerous predictions deviating beyond the $3\sigma$ boundary. This highlights its inability to capture critical timing dependencies due to the absence of physical characteristics in its modeling. Figure 8(b) shows that RTL-Timer improves upon MasterRTL, reducing scatter and aligning better with the

1 : 1 line. However, it still suffers from noticeable outliers, limiting its reliability in predicting TNS.

In Figure 8(c), RTLDistil without distillation achieves further improvements, with reduced scatter and better alignment with ground truth TNS values. However, some deviations persist, reflecting the challenges of accurate timing prediction without knowledge transfer. Finally, Figure 8(d) demonstrates that RTLDistil with distillation achieves the best alignment, with predictions tightly clustered along the 1 : 1 line and most points well within the $3\sigma$ boundary. This superior performance is attributed to the multi-granularity knowledge distillation framework, which transfers critical physical characteristics from the teacher model to the student model, enabling precise modeling of aggregate timing violations.

In conclusion, RTLDistil with distillation significantly outperforms prior approaches in TNS prediction, effectively bridging the abstraction gap between RTL and layout stages. By leveraging the forward-reverse propagation strategy and incorporating physical knowledge, RTLDistil enables highly accurate and reliable TNS predictions, critical for early-stage design optimization.

The correlation plots for AT, WNS, and TNS demonstrate the progressive improvements achieved by RTLDistil, particularly after incorporating the multi-granularity knowledge distillation process. RTLDistil with distillation consistently outperforms MasterRTL, RTL-Timer, and RTLDistil without distillation across all timing metrics. By effectively transferring physical characteristics from the teacher model to the student model, RTLDistil bridges the abstraction gap between RTL and layout stages, achieving superior timing prediction accuracy. These results highlight the practical applicability of RTLDistil in early-stage design optimization workflows.

## B. Appendix: Analysis of Forward and Reverse Propagation Strategies

In this appendix section, we analyze the impact of different propagation strategies on the performance of the RTL-level student model. Specifically, we investigate how varying the number of forward (F) and reverse (R) propagation passes affects the model's ability to predict timing-critical metrics, including Arrival Time (AT), Worst Negative Slack (WNS), and Total Negative Slack (TNS). This analysis is conducted on a set of challenging timing benchmarks, which were selected for their high complexity and difficulty. These benchmarks were used to train downstream tasks with the RTL model under various propagation configurations.

### B.1. Experimental Setup

We experimented with several combinations of forward and reverse propagation passes, including:

- 1 (Forward + Reverse): A single round of forward and reverse propagation.

- 2 (Forward + Reverse): Two rounds of forward and reverse propagation.

- 2 Forward: Two forward-only propagation passes.

- 5 (Forward + Reverse): Five rounds of forward and reverse propagation.

The experiments aim to demonstrate that: 1. A combination of forward and reverse propagation outperforms forward-only propagation by capturing bidirectional dependencies in the circuit graph. 2. Neither too few nor too many forward and reverse passes are optimal; instead, a balanced configuration achieves the best performance.

### B.2. Results and Discussion

The results of the experiments are summarized in Table 5. Key observations are as follows:

**Forward + Reverse vs. Forward-Only:** The results show that configurations with both forward and reverse propagation consistently outperform forward-only configurations. For instance, the "2 (Forward + Reverse)" model achieves higher Pearson Correlation Coefficients (PCC) and $R^2$ values across all metrics compared to the "2 Forward" model. Specifically, for WNS, the PCC improves from 0.8815 to 0.8870, and the Mean Absolute Percentage Error (MAPE) increases slightly from 33.21% to 36.87%. This demonstrates that reverse propagation effectively captures timing dependencies and a wider range of information about the surrounding circularity that forward propagation alone cannot.

**Optimal Number of Passes:** Among the configurations tested, "2 (Forward + Reverse)" achieves the best balance between performance and computational complexity. It achieves the highest PCC for AT (0.8446), the best $R^2$ for TNS

Table 5: Performance comparison of different propagation strategies for the RTL-level student model.

| Propagation Strategy | Arrival Time (AT) | | | Worst Negative Slack (WNS) | | | Total Negative Slack (TNS) | | |
|---|---|---|---|---|---|---|---|---|---|
| | PCC | $R^2$ | MAPE | PCC | $R^2$ | MAPE | PCC | $R^2$ | MAPE |
| 1 (Forward + Reverse) | 0.8328 | 0.6742 | **23.10%** | 0.8794 | 0.7599 | 34.30% | 0.8788 | 0.7685 | **36.38%** |
| 2 (Forward + Reverse) | **0.8446** | **0.6808** | 24.63% | **0.8870** | 0.7507 | 36.87% | **0.8875** | **0.7860** | 40.18% |
| 2 Forward | 0.8373 | 0.6805 | 24.80% | 0.8815 | **0.7601** | **33.21%** | 0.8801 | 0.7590 | 38.03% |
| 5 (Forward + Reverse) | 0.8325 | 0.6456 | 24.33% | 0.8820 | 0.7280 | 35.51% | 0.8786 | 0.7675 | 41.58% |

**Propagation Strategy**: Different combinations of forward (Forward) and reverse (Reverse) passes used during training and testing.

(0.7860), and competitive MAPE across all metrics. In contrast, "1 (Forward + Reverse)" underperforms due to insufficient propagation depth, while "5 (Forward + Reverse)" shows a slight degradation in performance, likely due to overfitting or noise amplification from excessive propagation. This finding highlights the importance of selecting an appropriate number of propagation passes.

**Impact of Over-Propagation:** The "5 (Forward + Reverse)" configuration demonstrates diminishing returns and even slight performance degradation compared to "2 (Forward + Reverse)." For example, the MAPE for TNS increases from 40.18% to 41.58%. This suggests that excessive propagation may introduce noise or overfit the model to spurious relationships in the graph, reducing its generalizability.

### B.3. Implications of Results

These findings provide strong evidence for the effectiveness of our propagation strategy. The combination of forward and reverse propagation enables richer contextual learning by incorporating bidirectional information flow, which is critical for accurately modeling timing dependencies in circuit graphs. Moreover, the experimental results validate our hypothesis that an optimal balance of propagation rounds is necessary for achieving the best performance. Too few passes fail to capture sufficient information, while too many passes may amplify noise or cause overfitting.

The results presented in this section reinforce the effectiveness of the proposed forward-reverse propagation strategy in the RTL-level student model. By carefully selecting the number of propagation passes, our approach achieves state-of-the-art performance on timing-critical tasks, demonstrating its suitability for challenging benchmarks with high timing complexity. These findings further substantiate the robustness and generalizability of our method for early-stage timing prediction in industrial workflows.

## C. Appendix: Expanded Mathematical Formulation of Knowledge Distillation

This appendix further extends the mathematical details of our cross-stage Knowledge Distillation (KD) process, emphasizing the core principles behind node-level, subgraph-level, and global-level alignment. We focus on equations and derivations with minimal textual explanation while ensuring clarity and correctness.

### C.1. Preliminaries and Notation

- $G^T = (V, E)$: Graph representing the circuit (layout) for the Teacher model, with $|V|$ nodes.

- $G^S = (V, E)$: Graph representing the circuit (RTL) for the Student model, with $|V|$ nodes.

- $\mathbf{z}_v^T, \mathbf{z}_v^S \in \mathbb{R}^{d_T}, \mathbb{R}^{d_S}$: Final-layer embeddings for node $v$ from Teacher and Student, respectively (typically $d_T > d_S$).

- $\mathbf{M} : \mathbb{R}^{d_S} \rightarrow \mathbb{R}^{d_T}$: Learnable alignment transform (e.g., an MLP) that maps Student embeddings to the Teacher's embedding space.

- $\| \cdot \|_p$: $p$-norm (often $p = 1$ or $p = 2$), used for measuring distance between vectors.

- $\mathrm{smoothL1}(\mathbf{x}, \mathbf{y})$: The smooth L1 (a.k.a. Huber) loss between vectors $x$ and $y$, defined as

$$\mathrm{smoothL1}(\mathbf{x}, \mathbf{y}) = \sum_{i=1}^{d} \begin{cases} \frac{1}{2}(x_i - y_i)^2, & \text{if } |x_i - y_i| < 1, \\ |x_i - y_i| - \frac{1}{2}, & \text{otherwise.} \end{cases} \quad (8)$$

This function behaves quadratically (like L2) near zero and transitions to a linear (L1) penalty for larger errors, thus often providing a more robust gradient profile than purely L1 or L2.

## C.2. Teacher *vs.* Student Outputs

We consider the Teacher model as having more rounds of forward-reverse propagation and seeing rich physical layout features. The Student sees only RTL features with fewer propagation steps:

**Teacher forward-reverse.** The forward propagation update for node $v$ at $(i+1)$-depth from input DFF node ($DPI$) in $r$-round propagation is formulated as:

$$\mathbf{h}_v^{(i+1),r} = \sigma\Big( \sum_{u \in \mathcal{N}(v)} \alpha_{vu}^{Forward} W^{Forward} \mathbf{h}_u^{(i),r} \Big), \quad i = 0, \ldots, \text{Level}_{DPI}, \tag{9}$$

where $\mathcal{N}(v)$ represents the forward neighboring nodes of $v$; $h$ denotes the feature vector of nodes; $\mathbf{W}$ denotes the linear transform weight and $\alpha$ denotes the coefficients calculated by the attention mechanism in (Velickovic et al., 2017). Similarly, the reverse propagation update for node $v$ at $(j+1)$-depth from output DFF node ($DPO$) in $r$-round is formulated as:

$$\mathbf{h}_v^{(j+1),r} = \sigma\Big( \sum_{u \in \mathcal{R}(v)} \alpha_{vu}^{Reverse} W^{Reverse} \mathbf{h}_u^{(j),r} \Big), \quad j = 0, \ldots, \text{Level}_{DPO}, \tag{10}$$

where $\mathcal{R}(v)$ denotes the reverse neighboring nodes of $v$. The teacher typically uses more rounds ($R_T$) to capture stronger physical effects.

$$\mathbf{h}_v^{\forall, R_T} \rightarrow \mathbf{z}_v^T, \tag{11}$$

**Student forward-reverse.** Student model share the same forward-reverse propagation with teacher model. The student model typically uses less rounds ($R_S$) to ensuring the reference efficiency.

$$\mathbf{h}_v^{\forall, R_S} \rightarrow \mathbf{z}_v^S \tag{12}$$

where $R_S \leq R_T$, reflecting fewer propagation rounds.

## C.3. Basic KD Approaches (Distribution-Based)

In classification-style KD, one often aligns teacher and student *soft targets* $\mathbf{p}^T$ and $\mathbf{p}^S$:

$$\mathbf{p}^T = \text{Softmax}\Big(\frac{\mathbf{z}^T}{\tau}\Big), \quad \mathbf{p}^S = \text{Softmax}\Big(\frac{\mathbf{z}^S}{\tau}\Big), \tag{13}$$

where $\tau > 0$ is the temperature. The classical KD objective is:

$$\mathcal{L}_{\text{KD,dist}} = \frac{\tau^2}{|V|} \sum_{v \in V} \text{KL}\Big(\mathbf{p}_v^T \,\|\, \mathbf{p}_v^S\Big), \tag{14}$$

$$\text{KL}(\mathbf{a}\|\mathbf{b}) = \sum_i a_i \log\Big(\frac{a_i}{b_i}\Big), \tag{15}$$

which encourages the Student distribution $\mathbf{p}^S$ to approximate $\mathbf{p}^T$.

However, in *regression-focused* timing prediction, we instead align continuous embeddings or real-valued outputs (e.g., arrival time). Below, we detail node-level, subgraph-level, and global-level alignment for *feature-based* KD.

## C.4. Feature-Based KD for Timing (Multi-Granularity)

### C.4.1. NODE-LEVEL FEATURE ALIGNMENT

Each node $v$ yields final embeddings $\mathbf{z}_v^T, \mathbf{z}_v^S$. To align dimensions, define:

$$\mathbf{z}_v^{S,\text{align}} = \mathbf{M}\big(\mathbf{z}_v^S\big). \tag{16}$$

Then a common choice is the $L_1$ or $L_2$ distance:

$$\mathcal{L}_{\text{node}} = \frac{1}{|V|} \sum_{v \in V} \left\| \mathbf{z}_v^T - \mathbf{z}_v^{S,\text{align}} \right\|_p. \tag{17}$$

This captures fine-grained alignment at each node (e.g., each register).

### C.4.2. SUBGRAPH-LEVEL FEATURE ALIGNMENT

For local contextual alignment, let $G(v)$ be $v$'s subgraph (e.g., fan-in cone). We pool Teacher embeddings:

$$Q^T(v) \;=\; \text{Pool}\big\{\mathbf{z}_u^T : u \in G^T(v)\big\}, \tag{18}$$

and similarly for the Student:

$$Q^S(v) \;=\; \text{Pool}\big\{\mathbf{z}_u^S : u \in G^S(v)\big\}. \tag{19}$$

Applying $\mathbf{M}$,

$$\mathcal{L}_{\text{sub}} = \frac{1}{|V|} \sum_{v \in V} \left\| Q^T(v) - \mathbf{M}\big(Q^S(v)\big) \right\|_p. \tag{20}$$

This enforces local structural insight in the Student.

### C.4.3. GLOBAL-LEVEL FEATURE ALIGNMENT

We also encourage global similarity:

$$\mathbf{g}^T = \text{Pool}\big\{\mathbf{z}_v^T : v \in V\big\}, \quad \mathbf{g}^S = \text{Pool}\big\{\mathbf{z}_v^S : v \in V\big\}, \tag{21}$$

leading to

$$\mathcal{L}_{\text{global}} = \left\| \mathbf{g}^T - \mathbf{M}\big(\mathbf{g}^S\big) \right\|_p. \tag{22}$$

This term addresses overall circuit-level properties (e.g., average load or total distribution shifts).

### C.5. Overall Distillation Objective

In addition to the above feature-based losses, we incorporate a supervised term, $\mathcal{L}_{\text{sup}}$, that aligns the Student's predicted timing metrics (e.g., arrival time $\widehat{\text{AT}}_v$) with reference labels $\text{AT}_v^T$ (Teacher or sign-off data). Hence,

$$\mathcal{L}_{\text{total}} = \mathcal{L}_{\text{sup}} \;+\; \alpha\,\mathcal{L}_{\text{node}} \;+\; \beta\,\mathcal{L}_{\text{sub}} \;+\; \gamma\,\mathcal{L}_{\text{global}}, \tag{23}$$

where $\alpha, \beta, \gamma$ weight the importance of each distillation granularity.

### C.6. Training-Testing Dynamics

**Teacher Training.** Using layout-based features and full forward-reverse passes, we minimize the Teacher's own supervised objective (e.g., MSE w.r.t. sign-off AT). Let $\mathbf{z}_v^T$ be the final teacher embeddings.

**Student Distillation.** We fix pretrained teacher embeddings $\{\mathbf{z}_v^T\}$ and then train our Student on the lighter RTL graph by jointly optimizing:

- $\mathcal{L}_{\text{sup}}$: e.g., L1/L2 between $\widehat{\text{AT}}_v^S$ and $\text{AT}_v^T$,

- $\mathcal{L}_{\text{node}}, \mathcal{L}_{\text{sub}}, \mathcal{L}_{\text{global}}$ (Equations (17), (20), (22), and we used $\text{smoothL1}$ for good measure.).

**Inference.** For a new RTL design, the learned Student GNN (with embedded distillation knowledge) performs limited forward-reverse propagation to estimate timing. Despite minimal overhead, this yields near-layout accuracy in arrival time, WNS, TNS, etc.

## C.7. Conclusion of KD Formulation

By combining distribution-based KD concepts with multi-granularity feature alignment, we robustly transfer high-precision physical knowledge from the Teacher to the Student. Equations (14)–(23) form the mathematical backbone of our cross-stage distillation and enable early-stage RTL timing prediction comparable to sign-off results.

## C.8. Fan-in Cone Construction and Mathematical Formulation

Fan-in cone construction plays a critical role in our multi-granularity knowledge distillation framework, particularly for subgraph-level alignment between RTL and layout stages. Here, we present a rigorous mathematical formulation of fan-in cone construction and its hierarchical representation.

### C.8.1. FORMAL DEFINITION OF FAN-IN CONE

**Definition C.1** (Fan-in Cone). Given a circuit graph $G = (V, E)$, where $V$ represents the set of nodes (including registers and combinational cells) and $E$ represents the set of directed edges. For any register node $v \in V$, its fan-in cone is defined as $G_{\text{fan-in}}(v) = (V_{\text{fan-in}}(v), E_{\text{fan-in}}(v))$, where:

$$V_{\text{fan-in}}(v) = \{u \in V \mid \text{exists a directed path from } u \text{ to } v\}, \tag{24}$$
$$E_{\text{fan-in}}(v) = \{(u, w) \in E \mid u, w \in V_{\text{fan-in}}(v)\}. \tag{25}$$

### C.8.2. FAN-IN CONE STRUCTURE

To facilitate effective knowledge transfer across different abstraction levels, we develop a level-based hierarchical fan-in cone representation:

**Definition C.2** (Fan-in cone Representation). The level of a node $u$ relative to target register $v$ is defined recursively as:

$$\text{Level}_{\text{fan-in}}(u, v) = \begin{cases} 0, & \text{if } u = v, \\ \max_{w:(u,w) \in E} \{\text{Level}_{\text{fan-in}}(w, v)\} + 1, & \text{if } u \text{ reaches } v, \\ \infty, & \text{otherwise.} \end{cases} \tag{26}$$

The nodes at each level $l$ (from the target register/DFF) are defined as:

$$V_l(v) = \{u \in V_{\text{fan-in}}(v) \mid \text{Level}_{\text{fan-in}}(u, v) = l\}. \tag{27}$$

The representation of the whole fan-in cone is defined as:

$$Q_{\text{fan-in}}(v) = \text{Pool}\{\mathbf{z}_u^T : u \in V_{\text{fan-in}}(v)\}, \tag{28}$$

where $\mathbf{h}_u$ represents final-layer embeddings for node $u$ in the fan-in cone, and we use mean pooling here.

The fan-in cone construction and its hierarchical representation serve as the foundation for our subgraph-level knowledge distillation, enabling the effective transfer of timing characteristics from the layout-level teacher to the RTL-level student model. Our experimental results demonstrate that this structured approach significantly improves the accuracy of timing prediction while maintaining computational efficiency.

