# OpenReview forum: "Bridging Layout and RTL: Knowledge Distillation based Timing Prediction"
_ICML.cc/2025/Conference — ICML 2025 spotlightposter_

### Official Review · Reviewer_wjHP · 2025-03-12

**Overall Recommendation:** 4

**Summary:**

This paper presents a cross-stage knowledge distillation framework for timing prediction. Under this framework, the layout aware teacher model distills layout charactersistics to an RTL-level student models. Experimental results demonstrate significant improvement compared with other prediction models.

**Claims And Evidence:**

The claims seem supported by experiments. Experiments on Table 3 clearly demonstrate the benefit of both distillation and finetuning, with distillation being more important. Table 4 further demonstrate importance of multi-granularity knowledge distillation.

**Essential References Not Discussed:**

None.

**Experimental Designs Or Analyses:**

Experiments seem solid. Ablation studies are well conducted.

**Methods And Evaluation Criteria:**

Benchmarks used seemed to be unclear. The authors claimed to use 2k RTL designs sourced from open-source projects. No further details are presented. Also remains unclear why not use similar data compared with open-source versions of prior work.

**Other Comments Or Suggestions:**

None.

**Other Strengths And Weaknesses:**

The authors idea of distilling layout aware models to RTL-level models is novel and effective. However the dataset and test used in the paper remains unclear. More comparision with prior work could also benefit the paper.

**Questions For Authors:**

(1) Can the authors clarify on the dataset used for training/testing? Why not direct use already open-source data from prior work. Will new data (and models) from authors be open-sourced?

(2) How did the authors conduct experiements with MasterRTL and RTL-Timer? Were they retrained on same data?

(3) Can the authors compare with more existing prior work other than from Fang?

**Relation To Broader Scientific Literature:**

The authors improve upon prior work by distilling layout-aware models to RTL code level models. Authors were able to present improvements compared with prior works.

**Theoretical Claims:**

The paper does not make any theoretical claims.

---

> ### Author Rebuttal · Authors · 2025-03-31
>
> **Dear Reviewer wjHP,**
>
> We sincerely appreciate the reviewer's positive remarks and valuable suggestions regarding our work, particularly concerning the sources of our benchmark data, the open-source plan, the fairness of comparisons with MasterRTL/RTL-Timer, and the discussion on comparison with other models. Below, we provide detailed answers to the points you raised.
> ## 1. Benchmark and Dataset Explanation
> We have carefully chosen not to rely exclusively on certain existing publicly available datasets (e.g., ISCAS-89, ITC-99) because they are limited in size and diversity relative to our specific RTL-Layout prediction goals. For some large circuits, such as some large-scale CPUs or other designs, there may be many duplicate or essentially identical paths in the circuit, but the timing analysis is more associated with paths, which makes the distribution of the existing datasets not particularly satisfying; we wanted to minimize the occurrence of duplicate paths in the datasets and to make the dataset distribution as even and diverse as possible in order to allow more physical information to be learned by our model.
>
> Instead, we collected more circuits of **different types, functions, and sizes** to reflect real-world industrial needs, including small arithmetic blocks, DSP modules, RISC-V subsystems, etc. This makes the prediction task harder, but we think it's more practical, pervasive, industrially valuable, and closer to the needs of actual industrial processes. These 2000+ designs come from GitHub, OpenCore, Hugging Face, and the open-source RISC-V project, ensuring broad functionality and design complexity coverage. The construction and cleaning of the dataset took us close to 5 months (see the reply to Reviewer DZ14 for more information on Layout data construction).
> ## 2. Open-Source Plan of Dataset and Codes
> Furthermore, we have partially anonymized and **released a subset of these designs and codes on public repositories (https://anonymous.4open.science/r/icml2025RTL-CBFD/)**; however, the size of back-end data and the space limitations on anonymous platforms prevent a complete release at this stage. We will expand our open dataset, aiming to foster community progress in AI4EDA.
>
> We will keep enhancing our open-source repository (both the dataset and the code) to facilitate replication and iterative improvement by the community. Thanks to its graph-oriented modeling and distillation-based learning, we strongly believe our framework can be generalized to more designs.
>
> ## 3. Comparisons with MasterRTL and RTL-Timer
> All results reported in the paper were derived by re-training and evaluating both MasterRTL [1] and RTL-Timer [2] on precisely the same dataset and evaluation protocols used by our method. Specifically:
>
> 1. Both are open-source implementations, and we reproduced them faithfully on our data, adhering to the authors' original procedures.
> 2. We used the **same train/validation/test partitioning and ground-truth labels** for all methods.
> 3. We evaluated using identical error metrics (MAPE, PCC, R²) for consistency.
> ## 4. Comparison with Other Models
> We appreciate your recommendation to compare more thoroughly with related methods. Given that MasterRTL and RTL-Timer are among the most competitive open-source approaches for RTL-level timing prediction, demonstrating improvements over them provides a fair and substantial benchmark. According to prior works (e.g., [3], [4]), they didn't show the same performance as MasterRTL and RTL-Timer (e.g., > 10% gap in MAPE). We chose MasterRTL and RTL-Timer as the strongest public baselines.
> ## Conclusion
> Thanks again for the reviewer's supportive review and for highlighting the significance of benchmark diversity, fair comparisons, and open-source work. We will refine the discussion on data sources, continue expanding our dataset for broader coverage, and work to open-source the code with the full dataset. We hope the clarifications and updates will further underscore the practicality and robustness of our method.
>
> *We greatly value the reviewer's feedback and look forward to any additional suggestions. Looking forward to getting a higher rating from you!*
> ## References
> [1] Fang, Wenji, et al. "Masterrtl: A pre-synthesis ppa estimation framework for any rtl design." *2023 IEEE/ACM International Conference on Computer Aided Design (ICCAD)*. IEEE, 2023.
>
> [2] Fang, Wenji, et al. "Annotating slack directly on your verilog: Fine-grained rtl timing evaluation for early optimization." *Proceedings of the 61st ACM/IEEE Design Automation Conference*. 2024.
>
> [3] Xu, Ceyu, Chris Kjellqvist, and Lisa Wu Wills. "SNS's not a synthesizer: a deep-learning-based synthesis predictor." *Proceedings of the 49th Annual International Symposium on Computer Architecture*. 2022.
>
> [4] Sengupta, Prianka, et al. "How good is your Verilog RTL code? A quick answer from machine learning." *Proceedings of the 41st IEEE/ACM International Conference on Computer-Aided Design*. 2022.

---

> > ### Comment · Reviewer_wjHP · 2025-04-01
> >
> > Thank you for the rebuttals on addressing the concerns regarding benchmark diversity. However, I do feel that incoporating more baselines would greatly strengthen the paper. It would also be interesting to see whether the proposed knowledge distillation approach would benefit other works/models such as [3,4].

---

> > > ### Author Response · Authors · 2025-04-08
> > >
> > > **Dear Reviewer wjHP,**
> > >
> > > Thanks for your thoughtful insights on incorporating additional baselines and investigating how our knowledge distillation (KD) framework might assist other models. We agree that broadening baseline comparisons and exploring the general applicability of our KD approach will enrich the ML for EDA community. Below, we offer details on additional baselines and applying our KD design within a path-based model like SNS.
> > > ### 1. Incorporating More Baselines
> > > As you point out, extending our experiments with additional baselines strengthens our work. We considered SNS [3] and similar approaches [4], but their unreleased code/data made fair reproduction difficult. Hence, we re-implemented SNS on our dataset and integrated our KD strategy into the re-implementation as closely as possible. Notably, SNS includes randomizing methods (e.g., DFS), causing performance random fluctuations; we thus provide one representative outcome.
> > >
> > > **SNS Performance (Without Distillation):**
> > > ```
> > >                      PCC     R²      MAPE
> > > Arrival Time (AT):   0.33   -0.76    83%
> > > WNS:                 0.68    0.45    70%
> > > TNS:                 0.41   -0.18    85%
> > > ```
> > > It shows that SNS's performance is considerably lower than RTLDistil's. The relatively poor performance is due to SNS employing a path-based approach, which struggles to capture complex physical properties (e.g., RC parasitics) and surrounding circuit information critical for accurate layout-level timing predictions.
> > > ### 2. Applying Knowledge Distillation to SNS
> > > By adapting our multi-granularity KD to the path-based SNS model, we implemented a partial version of subgraph- and global-level, but did not implement node-level due to the principles of the model itself:
> > >
> > > 1. **Node-Level Distillation**
> > >    - *Challenge:* SNS encodes circuits as sampled paths using a lightweight Transformer. Registers appear across multiple paths, lacking a unified per-register embedding. Thus, straightforward one-to-one node distillation is hardly feasible.
> > > 2. **Subgraph-Level Distillation**
> > >    - *Feasibility:* Paths sharing the same sink or source register can be grouped to approximate local cones. We create a subgraph embedding by aggregating the path Transformers' hidden states and align these embeddings with the teacher's subgraph outputs.
> > > 3. **Global-Level Distillation**
> > >    - *Straightforward:* SNS produces a global design-level prediction, making applying a global-level Smooth L1 loss between teacher and student representations simpler.
> > >
> > > **SNS Performance (With Distillation):**
> > > ```
> > >                      PCC     R²      MAPE
> > > Arrival Time (AT):   0.71    0.52    41%
> > > WNS:                 0.73    0.68    53%
> > > TNS:                 0.81    0.70    55%
> > > ```
> > > This significantly improves SNS, yet remains below RTLDistil due to fundamental architectural constraints. It exhibits limitations in single-point AT prediction, possibly due to the lack of good node-level distillation. The path-based approach inherently struggles to capture the surrounding circuit context, critical for layout-level timing prediction.
> > > ### 3. Inherent Constraints of Path-Based Methods
> > > Our findings underscore how path-based algorithms, typified by SNS, face inherent challenges in capturing complex layout- and design-level interactions:
> > > - **Limited Circuit Context:** Accurate layout timing is determined by a range of geometrical and parasitic factors that extend beyond single paths.
> > > - **Physical Parameter Integration:** Resistance, capacitance, coupling, and congestion effects must be comprehensively integrated. Path-based sampling often oversimplifies these interdependencies.
> > > - **Long-Distance Dependence:** Surrounding circuit information outside the direct path significantly affects timing accuracy. Path-based methods often find it difficult to capture these long-term dependencies.
> > >
> > > By contrast, our GNN-based RTLDistil framework, combined with the domain-specific asynchronous forward-reverse propagation, captures entire local and global contexts, thus embedding physical knowledge more holistically into the distilled student model.
> > > ### 4. Concluding Remarks
> > > We appreciate your encouragement to explore broader baselines and experiment with KD in other modeling paradigms. These endeavors confirm that our approach is both *general*—improving other frameworks—and *powerful* when used in a fully GNN-based setting. This further highlights KD's strong potential to speed up and refine EDA tasks, especially under the "left-shift" paradigm.
> > >
> > > Moving forward, we will:
> > > - Refine our multi-granularity KD framework for potential collaboration with more EDA problems.
> > > - Expand our open-source resources to facilitate reproducibility and development.
> > >
> > > Your deep consideration clarifies how our KD approach can influence broader ML and EDA challenges.
> > >
> > > *Thanks again for your time and valuable feedback. We hope these additional experiments and explanations address your concerns. We very much hope to earn your affirmation and a higher rating!*

---

### Official Review · Reviewer_55gZ · 2025-03-17

**Overall Recommendation:** 3

**Summary:**

The paper proposes RTLDistil, a framework aimed at bridging the gap between early-stage RTL timing prediction and accurate layout-level timing analysis. The method leverages a dual-model setup: a high-capacity teacher model operating on a layout netlist graph and a lightweight student model working on an RTL-level Simple Operator Graph (SOG). The core idea is to use multi-granularity knowledge distillation that encompasses node-, subgraph-, and global-level alignments, along with an asynchronous forward-reverse propagation strategy to transfer precise physical characteristics from the teacher to the student. The authors claim that RTLDistil reduces prediction errors compared to previous methods such as MasterRTL and RTL-Timer.

**Claims And Evidence:**

The claims in the paper are generally well-supported by corresponding analyses and experiments. Building on this, I have the following question:

- To facilitate an effective "left-shift" in the design process, it would be beneficial to compare the results with analytical STA methods in terms of computational efficiency. This comparison would provide readers with a clearer understanding of the achieved accuracy-efficiency trade-off.

**Essential References Not Discussed:**

It would be helpful to add a detailed discussion on the similarities and differences in terms of method and results of many closely related papers on the same topic, such as:

[1] Moravej, Reza, et al. "The Graph's Apprentice: Teaching an LLM Low Level Knowledge for Circuit Quality Estimation." arXiv preprint arXiv:2411.00843 (2024).

[2] Zhong, Ruizhe, et al. "Preroutgnn for timing prediction with order preserving partition: Global circuit pre-training, local delay learning and attentional cell modeling." Proceedings of the AAAI Conference on Artificial Intelligence. Vol. 38. No. 15. 2024.

[3] DynamicRTL: RTL Representation Learning for Dynamic Circuit Behavior

**Experimental Designs Or Analyses:**

- The performance achieved by the teacher model still appears relatively limited. A more in-depth analysis of the teacher model’s quality and the current limitations of the student model’s performance would be beneficial. It would be helpful to clarify whether these limitations stem from the teacher model’s capability, the learning process, or other factors.

- How are the losses across the three granularities balanced? Is this parameter sensitive to performance? What is the computational cost of a grid search? Additionally, is the grid search a one-time process, or does it need to be adjusted for different models, tasks, or datasets?

**Methods And Evaluation Criteria:**

Overall, I believe the proposed method is a valuable exploration in bridging the gap between early-stage RTL timing prediction and accurate layout-level timing analysis. The approach appears well-reasoned. However, I have the following concerns regarding its technical contributions:

- Forward-Reverse Propagation Strategy: The bidirectional propagation mechanism, which aggregates fan-in and fan-out information, closely resembles standard bidirectional message-passing techniques widely used in GNNs for RTL [1]. The asynchronous adaptation seems like a minor modification, making the technical contribution of this aspect relatively limited.

- Multi-Granularity Alignment: Aligning features at the node, subgraph, and global levels is a common technique in GNNs, as seen in [2]. While its integration within a knowledge distillation framework tailored for EDA is interesting, it represents an incremental improvement rather than a fundamentally novel contribution.

[1] Lopera, Daniela Sánchez, and Wolfgang Ecker. "Applying GNNs to timing estimation at RTL." Proceedings of the 41st IEEE/ACM International Conference on Computer-Aided Design. 2022.

[2] Zhang, Muhan, and Pan Li. "Nested graph neural networks." Advances in Neural Information Processing Systems 34 (2021): 15734-15747.

**Other Comments Or Suggestions:**

Please refer to the above sections.

**Other Strengths And Weaknesses:**

Please refer to the above sections.

**Questions For Authors:**

- Although the paper demonstrates improvements over baseline approaches, to what extent do the current results meaningfully enable the claimed 'left-shift' of the design process? A more detailed analysis of the achieved results, supported by relevant references, would be beneficial.

- This may be an ambitious question, but I am curious about how RTLDistil performs on large-scale industrial designs (e.g., SoCs with millions of gates). The experiments primarily focus on medium-sized designs, yet scalability to larger designs is crucial for practical adoption.

- In Table 4, why do different distillation objectives lead to varying results for RTLDistil? Specifically, why does the w/ Node objective yield a promising MAPE while other metrics show less favorable outcomes? Are there insights into the underlying distribution of the achieved results?

**Relation To Broader Scientific Literature:**

N/A

**Theoretical Claims:**

Not applicable.

---

> ### Author Rebuttal · Authors · 2025-03-31
>
> **Dear Reviewer 55gZ,**
>
> Thanks for your insightful comments. Your feedback greatly helped us refine our work. Below, we address your major concerns.
> ## 1. Efficiency vs. STA and Left-Shift
> The entire flow from RTL to Layout to STA completion usually takes hours to days under rough quantitative estimation. Our method achieves several orders of magnitude faster runtime (see our response to Reviewer DZ14 for runtime details), with PCC 0.92, R² 0.85, and MAPE 16.87% for AT prediction—sufficient for early-stage RTL optimization [1], allowing the design flow to shift left.
> ## 2. Domain-specific Asynchronous Forward-Reverse Propagation
> While bidirectional GNNs exist, we focus on an asynchronous forward-reverse approach tailored to timing semantics and physical contexts. This provides a deeper coupling between AI modeling and timing prediction domain-specific requirements:
> - We incorporate multi-round, asynchronous updates to capture the long-range dependencies of registers, cells, and wires.
> - This schema includes a form of "reverse constraint" flow, reflecting the physical insight that timing slack on downstream cells can impose constraints on upstream segments, especially under realistic RC conditions.
> - Ablation results (Appendix) confirm simpler forward-only or fewer propagation rounds significantly degrade accuracy. This is caused by insufficient access to information about the surrounding circuit.
>
> Our design goes beyond standard message passing to encode domain-specific timing semantics.
> ## 3. Multi-Granularity Distillation: Tailored to Circuit Structures
> Our contribution lies in adapting multi-granularity distillation to **EDA-specific adaptation**:
> - **Tailoring to timing.** Each granularity naturally corresponds to specific circuit structures relevant to timing: node-level for register endpoints, subgraph-level for critical local paths (fan-in/out cones), and global-level for capturing overall congestion (e.g., WNS/TNS).
> - **Simultaneous cross-stage distillation.** By tuning the student at three granularities during distillation under the teacher, we alleviate the lack of physical cues at RTL by implicitly embedding critical layout-level physical information into the RTL GNN while focusing on the circuit's local and overall situation. As reviewer said, different distillation goals exhibit different behaviors because each of the three granularities plays a different role.  As in Table 4, removing any one or two granularities reduces performance, confirming their complementary roles.
>
> W/ Node performs well on AT's MAPE because it ignores the global or several paths in relation to each other and focuses only on registers. This leads to relatively good predictions in numerical values (MAPE) but weaker performance in correlation metrics (PCC,  R²) than the full model, not giving the best results.
> ## 4. Teacher Model Limits and Distillation Balance
> Our layout-level teacher is more accurate than the student but less precise than full STA due to the complexity of real RC extraction and GNN inherent limitations. The teacher’s precision naturally binds the student.
>
> Regarding grid search of distillation-loss weights (α, β, γ), we observed small fluctuations across different metrics. We believe the optimal balance varies with data properties, circuit complexity, and task objective focus. Via coarse-grained grid search (lower computational overhead), we find equal weights (α = β = γ) that show superior multi-task average performance and make the ablation experiments clearer.
> ## 5. Distinction from Related Work
> Compared to prior methods:
> - [2] uses LLMs but lacks scalability and physical layout fidelity.
> - [3] operates at layout stage, limiting early RTL optimization.
> - [4] targets dynamic runtime behavior, not static timing.
>
> Our method uniquely shifts accurate physical timing prediction to the RTL stage via cross-stage distillation.
> ## 6. Scalability to Industrial Designs
> We are scaling to large designs (e.g., BOOM). While back-end optimization of million-gate designs is very demanding regarding runtime (more than days) and hardware resources, our graph-based architecture has shown scalability in principle, and we will endeavor to report results as large circuits are available.
> ## Conclusion
> Thanks for your constructive input, which helped us better articulate our approach's novelty and practical utility.
>
> *We appreciate your time and look forward to further suggestions. Hope to get a higher rating from you!*
> ## References
> [1] Fang, Wenji, et al. "Annotating slack directly on your verilog: Fine-grained rtl timing evaluation for early optimization."
>
> [2] Moravej, Reza, et al. "The Graph's Apprentice: Teaching an LLM Low Level Knowledge for Circuit Quality Estimation."
>
> [3] Zhong, Ruizhe, et al. "Preroutgnn for timing prediction with order preserving partition: Global circuit pre-training, local delay learning and attentional cell modeling."
>
> [4] DynamicRTL: RTL Representation Learning for Dynamic Circuit Behavior

---

### Official Review · Reviewer_DZ14 · 2025-03-18

**Overall Recommendation:** 4

**Summary:**

The paper describes a multi-level distillation framework to train a tool that can predict final timing characteristics of a synthesis flow from RTL-level description.  The paper provides interesting detailed analysis of their results and argues that their results are much more accurate than prior efforts, many of which stop at an intermediate "gate-level" presentation.

**Claims And Evidence:**

The paper argues their accuracy is much better than SOTA algorithms on a range of circuits.  They argue and use ablation studies to evaluate the benefits of their multi-hierarhical approach.  Their reported numbers are indeed impressive and their approach IMHO has value. However, I think the evidence is not convincing. In particular, the results from final layout of a circuit depend on a myriad of specific parameters to the specific place and route tool and defined characteristics of the clock tree and whether or not certain optimizations are set or not (e.g., joint clock and data optimization), which can yield a run-time vs performance trade-off. A small change in initial area utilization during placement, for example, can yield quite significant differences post place-and-route. The description of the experiments do not describe these tool parameters in any detail and this reviewer is left to assume that the parameters are left the same between training and evaluation. This suggests that the model they train may only good be good for predicting timing for this specific set of tool parameters.  Given experts often tweak these parameters from design to design, this issue should at the very least be discussed. I have no doubt that their approach has the potential benefits they claim, but I think the actual numbers they provide should be explained and the limitations of their approach should be more fully described..

The paper also claims that the models are computationally efficient, but do not provide any run-time analysis of the models. This should be added.

**Essential References Not Discussed:**

None.

**Experimental Designs Or Analyses:**

See above.

**Methods And Evaluation Criteria:**

The set of benchmark circuits is only vaguely designed. In most EDA/CAD papers the results of each benchmark should be included and this is missing from this paper. I would have expected to see this in an anonymous git or a table in the appendix. Not sure why this is not here, but it makes re-producing these results quite difficult.

**Other Comments Or Suggestions:**

None.  My score has been adjusted to account for the very nice rebuttal the authors wrote. Thanks for the clarification.

**Other Strengths And Weaknesses:**

None.

**Questions For Authors:**

Please address the questions I raised above.

**Relation To Broader Scientific Literature:**

This paper is fundamentally about using ML for EDA and in that domain seems like reasonable work. It is not clear how useful or interesting it will be to the ICML community, but I am reviewing it as if it is.

**Theoretical Claims:**

There are no theoretical claims in the paper.

---

> ### Author Rebuttal · Authors · 2025-03-30
>
> **Dear Reviewer DZ14,**
>
> Thank you for your constructive comments and recognition of our work's core contributions. We greatly appreciate your thoughtful feedback, which has helped us further refine and clarify several key aspects of our methodology. Below, we provide a structured response to your concerns.
> ## 1. Multiple EDA Back-end Parameters for Iterative Design Optimization
> In constructing our dataset, we designed a unified, fully-automated back-end flow using state-of-the-art commercial tools—Synopsys Design Compiler and Cadence Innovus—with a consistent set of optimization switches (e.g., gate sizing, buffer insertion, cell movement, etc.). However, the circuits in our dataset were **not finalized under a single fixed configuration**.
>
> For each design and each back-end optimization-related parameter (e.g., density thresholds, routing constraints, clock constraints), we **automatically tried multiple sets of values and iteratively explored multiple different configurations**, often conducting **tens of design runs**, until the circuit reached a state where:
> - Placement density no longer increased, and
> - Timing metrics converged stably through repeated optimization.
>
> This convergence point is a practical proxy for physical design quality, reflecting an optimization level comparable to manually refined industrial flows. By doing so, we avoid biasing our dataset toward a singular "super-convergent" setting and instead generate diverse yet quality layouts more representative of industrial standards.
>
> Importantly, this means our model is **not tuned to predict timing under a specific tool configuration**, but rather aims to approximate the **best achievable timing performance** after realistic optimization—an objective more aligned with industrial design goals.
> ## 2. Industrial Relevance and Methodological Scalability
> We acknowledge that experienced experts in the industry may pursue further manual tuning to push timing closure closer to its theoretical optimum. However, given our goal of building a large RTL-to-Layout database covering many diverse designs, it is tough to explore every possible EDA configuration or foundry setting exhaustively.
>
> While our approach may not guarantee absolute optimality for each circuit, it reflects a **robust and converged implementation quality**, providing a meaningful basis for our timing prediction framework. By generating datasets by "trying multiple parameters iterations until convergence", our multi-granularity cross-stage distillation framework maintains strong generality across various designs, process nodes, and EDA flows. The student model is designed to be easily extensible—new designs, tools, or manufacturing conditions can be seamlessly integrated. Rather than overfitting to a specific toolchain or configuration, our framework demonstrates that cross-stage learning is broadly applicable and practical. Once more finely tuned industrial data become available, our method can directly incorporate such data to enhance model accuracy further.
> ## 3. Runtime and Computational Efficiency
> Performing the full RTL-to-Layout timing analysis flow—including synthesis, place-and-route, STA, etc.—typically consumes **hours or even days** for large circuits. In contrast, our GNN-based predictor operates at the RTL level, completing end-to-end inference in **seconds to a few minutes** for circuits ranging in size from thousands to millions of gates.
>
> For example, on a 400K-gate CPU design running on an A100 GPU server:
> - Logic synthesis takes ~30 minutes
> - Backend optimization (e.g. place-and-route, etc.) takes over 6 hours
> - STA requires ~33 minutes
>
> By comparison:
> - SOG extraction: ~65 seconds
> - Final GNN inference: <100 seconds
>
> This enables a **speedup of over 10×**, even against standalone STA, and far more when compared to the entire back-end flow. Moreover, our GNN model supports **parallel processing**, enabling efficient handling of large-scale circuits and making it viable for early design-stage performance estimation.
> ## 4. Data Availability and Reproducibility
> To support reproducibility and peer evaluation, we have anonymized and **released a subset of our code and data through an anonymous repository (https://anonymous.4open.science/r/icml2025RTL-CBFD/)**.
>
> Due to the large file sizes and platform storage limitations, only a partial dataset is available at this stage (a clearer description of the data aspects is included in the reply to Reviewer wjHP; please check it out). We intend to release the **full dataset and code** upon paper acceptance, ensuring transparency and enabling broader research engagement.
> ## Conclusion
> Thanks for your thoughtful feedback. Your comments have helped us improve the clarity of our presentation, particularly around our data collection methodology, the generalizability of the proposed framework, and its practical runtime advantages.
>
> *We sincerely appreciate your review and very much hope to earn your affirmation and a higher rating!*

---

### Decision · Program_Chairs · 2025-05-01

**Decision:**

Accept (spotlight poster)

**Comment:**

This paper presents RTLDistil, a cross-stage knowledge distillation framework that enables accurate timing prediction at the RTL level by leveraging layout-aware GNN-based teacher models. The key contribution lies in transferring physical characteristics from the layout domain into the early-stage RTL domain, using a multi-granularity distillation strategy (node-, subgraph-, and global-level) and an asynchronous forward-reverse message-passing scheme that captures timing-critical constraints more effectively than conventional GNNs. Empirical results show significant improvements in prediction metrics over strong baselines like MasterRTL and RTL-Timer, suggesting RTLDistil enhances both accuracy and efficiency for early design-stage optimisation.

While some reviewers noted that components like multi-granularity alignment and bidirectional propagation resemble existing techniques, the authors effectively clarified that their asynchronous design was tailored to timing semantics and physical propagation constraints unique to EDA tasks. They also addressed concerns about benchmark construction, reproducibility, and scalability to industrial designs through detailed responses and by releasing anonymized datasets and code. Additional experiments showed the generalizability of their KD framework when adapted to other models such as SNS, highlighting both its benefits and the challenges of path-based approaches.

Overall, the method offers a practical and scalable solution to bridging layout-level precision and RTL-level efficiency in timing prediction. The combination of robust experimental results, thoughtful design choices tailored to EDA, and clear evidence of broader applicability makes this work a valuable contribution to AI4EDA. Therefore, I recommend acceptance.